# Influence of Crosslinking Methods on Biomimetically Mineralized Collagen Matrices for Bone-like Biomaterials

**DOI:** 10.3390/polym15091981

**Published:** 2023-04-22

**Authors:** Jeremy Elias, Bobbi-Ann Matheson, Laurie Gower

**Affiliations:** Department of Materials Science & Engineering, University of Florida, Gainesville, FL 32611, USA; jelias@ufl.edu (J.E.); bobbimatheson99@gmail.com (B.-A.M.)

**Keywords:** biomineralization, biomimetic processing, PILP, collagen mineralization, crosslinking

## Abstract

To assist in bone defect repair, ideal bone regeneration scaffolds should exhibit good osteoconductivity and osteoinductivity, but for load-bearing applications, they should also have mechanical properties that emulate those of native bone. The use of biomimetic processing methods for the mineralization of collagen fibrils has resulted in interpenetrating composites that mimic the nanostructure of native bone; however, closely matching the mechanical properties of bone on a larger scale is something that is still yet to be achieved. In this study, four different collagen crosslinking methods (EDC-NHS, quercetin, methacrylated collagen, and riboflavin) are compared and combined with biomimetic mineralization via the polymer-induced liquid-precursor (PILP) process, to obtain bone-like collagen-hydroxyapatite composites. Densified fibrillar collagen scaffolds were fabricated, crosslinked, and biomimetically mineralized using the PILP process, and the effect of each crosslinking method on the degree of mineralization, tensile strength, and modulus of the mineralized scaffolds were analyzed and compared. Improved modulus and tensile strength values were obtained using EDC-NHS and riboflavin crosslinking methods, while quercetin and methacrylated collagen resulted in little to no increase in mechanical properties. Decreased mineral contents appear to be necessary for retaining tensile strength, suggesting that mineral content should be kept below a percolation threshold to optimize properties of these interpenetrating nanocomposites. This work supports the premise that a combination of collagen crosslinking and biomimetic mineralization methods may provide solutions for fabricating robust bone-like composites on a larger scale.

## 1. Introduction

Over 500,000 people in the US receive treatment for bone defects each year, a number that is expected to grow with an aging US and global population [1]. For critical-size defects requiring bone grafts, substitute materials that mimic the structure and properties of native bone are desired for creating the next generation bone substitute, that can be both load-bearing as well as bioresorbable. Biomimetic processing methods have proven useful for recreating the complex nano- and microstructure of native bone [2,3,4,5], providing materials that are promising for bone tissue regeneration [2,6,7]. Our group’s efforts in mimicking both the composition and nanostructure of native bone provide a significant step toward the goal of making bone substitutes that are not simply resorbable through hydrolytic or enzymatic degradation, but could potentially be remodeled via cellular control through the bone remodeling unit (BRU) [8], as occurs in native bone.

Previous studies by our group have created bone-like collagen-calcium phosphate composites, through intrafibrillar mineralization of collagen matrices, where charged soluble polymers are used as PILP process-directing agents [9] to sequester and stabilize amorphous calcium phosphate precursors that are able to infiltrate the fibrils [2]. This results in an interpenetrating composite that is remarkably similar to the nanostructure of native bone [2,10], contrasting with conventional methods of collagen mineralization (such as soaking in SBF types of solutions), which generally yield extrafibrillar deposits on collagen surfaces. Intrafibrillar mineralization has also been shown to contribute to properties such as modulus and hardness in the dentin structure, which is analogous to bone as a mineralized collagenous tissue [11]. Even though these interpenetrating collagen–mineral composites are similar in composition and structure to native bone, the mechanical properties, such as tensile strength and modulus, of many mineralized matrices still fail to match the properties of secondary bone, which are desirable for preparing load-bearing bone tissue replacements. Notably, similar mechanical properties to primary (woven) bone have been achieved, in this case by using compressive approaches to densify isotropic collagen matrices [9]. This densification approach will be used here as a quick means to screen various crosslinkers.

In native bone, the collagen matrix is stabilized through a variety of crosslinks, such as lysyl oxidase enzymatic crosslinking, created during bone formation [12]. This stabilization of the bone ECM through physiological crosslinking has been shown to improve the strength and modulus of native bone in both in vivo [13] and in vitro [14] studies. Given the complexity of enzymatic crosslinking found in biological tissues, which is controlled both spatially and temporally during matrix deposition, and differs amongst the different types of collagenous tissues, it is difficult to design a fully biomimetic method for crosslinking collagen matrices. Therefore, for collagen-based constructs (particularly for “soft” tissue engineering scaffolds), physical and chemical crosslinking methods are commonly utilized to tailor the strength and degradation properties of these natural biopolymers to match the properties of host tissue. Earlier crosslinking methods, such as glutaraldehyde (GTA) [15] and formaldehyde [16], although possessing the ability to enhance the mechanical properties, suffered from biocompatibility problems associated with their degradation products. Therefore, in recent years, a prominent method of crosslinking collagen for this purpose is carbodiimide chemistry. Crosslinking methods, such as 1-ethyl-3-(3-dimethylaminopropyl) carbodiimide (EDC) coupled with *N*-hydroxysuccinimide (NHS) [17], carry the advantage of being zero-length crosslinkers, improving biomechanical properties without the toxic degradation products of earlier methods. The ability to decrease degradation rates and increase the modulus of constructs, even at low concentrations, is significant, because the formation of fewer crosslinks uses up less of the cell-interactive amine and carboxylate groups on collagen residues, and keeping these groups intact preserves processes such as integrin-mediated binding [18]. Cytotoxic effects have also been mitigated using non-toxic nature-derived agents, such as genipin [19] and quercetin [20]. Genipin and quercetin provide the benefit of favorable cell compatibility [21], with quercetin serving as a possible cost-effective and biodegradable alternative to genipin [20]. Photocrosslinking methods, such as the use of riboflavin as a photoinitiator in the presence of UV light [22,23,24], or the modification of collagen with methacrylate groups [25,26], have also been utilized to tailor and improve the mechanical properties of collagen constructs, with the additional advantage of allowing the collagen matrices to be chemically altered in the presence of cells. While these methods have shown enhanced mechanical properties of “soft” collagen constructs (such as for tissue engineering applications), the effects of many of these crosslinking methods, when combined with biomimetic intrafibrillar mineralization processes for “hard” tissue constructs (for bone or dental tissue regeneration), have not been fully investigated. Therefore, our goal here was to compare these four diverse crosslinking methods, to evaluate if they alter the ability to achieve intrafibrillar mineralization via the PILP process, which in turn might affect the structure and thus mechanical properties of these biomimetically mineralized constructs. The chemical structures of the four crosslinkers are shown in Appendix A (see in Appendix A).

In the case of PILP-mineralized collagen, one study found that crosslinking can affect the degree of mineralization and properties in a favorable manner [27]. However, in our study on remineralization of slices of demineralized bone, the remineralization capability varied depending on the region of bone, with the newer osteonal regions being more readily mineralized [28]. At that time, we speculated that the age-related crosslinking of the interosteonal regions may have been more inhibitory to the mineralization. With these conflicting effects of crosslinking, it seems clear that it could be dependent on the type of crosslinking, and thus each type of crosslinker may require further study.

In the study reported herein, densified disorganized scaffolds have been prepared, which allow for a relatively rapid fabrication of fibrillar collagen matrices that display some similarities to woven bone [27]. Bovine type I collagen was utilized in this study because of its availability and similarity to human collagen [29]. Although these scaffolds lack the parallel-fibered order of dense lamellar bone and thus possess some porosity, they allow for easier and less expensive preparation of larger-scale samples, as needed for tensile testing to assess the large variety of crosslinking conditions. We anticipate that the information obtained from the crosslinking effects on these densified disordered matrices can then be applied to the fully dense, ordered scaffolds previously prepared by our group using the ‘molecular crowding’ approach to more closely mimic dense lamellar bone [30,31].

For matrix mineralization, crosslinked and non-crosslinked collagen matrices were mineralized via the PILP process, mediated by osteopontin (OPN). OPN is an intrinsically disordered protein, one of the several small integrin-binding ligands, *N*-linked glycoproteins (SIBLING), that constitute the non-collagenous proteins (NCPs) involved in bone formation [32,33,34]. We use the more readily available and inexpensive form, extracted from bovine milk (even though it is thought to be more heavily phosphorylated than bone OPN) [35], because it leads to rapid and uniform intrafibrillar mineralization [8,32], thereby facilitating the assessment of crosslinking effects on both the mineralization process and the properties of the formed matrices.

The effects of chemical crosslinking of type I collagen via EDC/NHS and quercetin, as well as the UV-mediated crosslinking of methacrylated collagen and riboflavin-incorporated collagen, were evaluated with respect to their impact on the OPN-mediated PILP mineralization process. Scanning electron microscopy (SEM) imaging was performed for analysis of the microstructure of mineralized collagen fibrils (fibril size, uniformity, and morphology), and combined with energy dispersive spectroscopy (EDS) to confirm the presence of intrafibrillar mineral in the scaffolds. Thermogravimetric analysis (TGA) was also performed, to quantitatively assess the mineral content of the scaffolds with respect to crosslinking levels for each respective method. Because of the impact of collagen crosslinking and matrix stabilization on a variety of properties, both in synthetic scaffolds and native bone, tensile testing was chosen to probe the bulk modulus and tensile strength in response to crosslinking of our fabricated matrices.

## 2. Materials and Methods

### 2.1. Collagen Scaffold Densification and Preparation

Collagen scaffolds, in the form of dense films, were prepared from type I collagen (TeloCol^®^, Advanced Biomatrix, Carlsbad, CA, USA). An acetic acid solution of type I collagen monomers (tropocollagen triple helical units), at a concentration of 3 mg/mL, was brought to neutral pH using 0.1 M NaOH and inserted into dialysis cassettes (Slide-A-Lyzer™ 3.5k MWCO, ThermoFisher Scientific, Waltham, MA, USA). The collagen was incubated at 37 °C for 4 h to induce fibrillogenesis and thus polymerize the scaffold, and then the cassette was transferred to a 500 mL solution of polyethylene glycol (PEG, 35,000 MW, 40% *w*/*v*., Sigma-Aldrich, Darmstadt, Germany) and left for 20 h, to densify the collagen. The densified collagen scaffolds in the dialysis cassette were then transferred to a 1X phosphate buffered saline (PBS) solution and incubated at 37 °C for 2 h, to rinse. The samples were then cut and peeled from the dialysis cassette and cut into 2 cm × 0.5 cm rectangular strips, with an approximate thickness of 100 µm. Samples were either immediately crosslinked or lyophilized to store for further testing.

For the photocrosslinked collagens (methacrylated collagen and riboflavin-incorporated collagen), collagen samples were prepared using the same method as above except for the initial premixing, as follows: For riboflavin-incorporated collagen, riboflavin 5′-phosphate sodium salt hydrate (Sigma-Aldrich) was added to the neutralized TeloCol^®^ to a concentration of 0.01 *w*/*v*% before insertion into dialysis cassettes. Riboflavin was also added to the PEG solution at a concentration of 0.01 *w*/*v*%, to prevent loss of riboflavin from the collagen gel during densification. For methacrylated collagen scaffolds, methacrylated collagen (PhotoCol^®^, Advanced Biomatrix) was brought to a concentration of 3 mg/mL and a neutral pH through the addition of NaOH. The photoinitiator (Irgacure 2959, Advanced Biomatrix), dissolved in neat methanol at 10% *w*/*v* concentration, was mixed with the collagen solution before insertion into the dialysis cassettes for densification.

### 2.2. Collagen Crosslinking

EDC/NHS. Carbodiimide crosslinking was carried out in a solution of 75% ethanol with 50 mM 2-(*N*-morpholino)ethanesulfonic acid (MES) hydrate (Sigma-Aldrich). A mass ratio of EDC:NHS:collagen of 1.15:0.276:1.0 was designated as 100% according to previous literature [17]. EDC and NHS were added to the ethanol solution to obtain 50%, 100%, and 200% solutions relative to the weight of the collagen scaffolds. The collagen scaffolds (2 cm × 0.5 cm strips) were immersed in their respective solutions and incubated at room temperature for 20 h on a shaker plate.

Quercetin. Quercetin crosslinking was carried out in a 40% ethanol solution. Quercetin dihydrate (Alfa Aesar, Haverhill, MA, USA) was first added to a 4 mL solution of 100% ethanol to achieve solubilization of the quercetin, and water was then added dropwise to obtain a 10 mL solution of quercetin in 40% ethanol. Quercetin solutions were prepared with ratios of 50, 100, and 200 wt% relative to the collagen scaffolds. Scaffolds were incubated in the quercetin solutions for 4 h and then rinsed with DI H_2_O (stirred in 500 mL for 45 min, with DI H_2_O replaced every 15 min).

UV Crosslinking. Crosslinking of methacrylated and riboflavin-incorporated collagen scaffolds was performed under 365 nm UV light immediately after removal from the dialysis cassettes. The methacrylated collagen scaffolds were crosslinked for irradiation times of 45, 90, and 300 s, and the riboflavin-incorporated collagen scaffolds were crosslinked for irradiation times of 60 and 300 s.

### 2.3. Collagen Mineralization

Solutions of 9 mM calcium chloride (CaCl_2_·2H_2_O, Fisher Scientific) and 4.2 mM potassium phosphate (K_2_HPO_4_, Fisher Scientific) were prepared in Tris-buffered saline (TBS) containing 0.9% (*w*/*v*) NaCl (Fisher Scientific) and 0.02% (*w*/*v*) sodium azide (Sigma-Aldrich). Osteopontin (OPN) mix derived from bovine milk (Lacprodan^®^ OPN-10, from Arla Food Ingredients Group P/S (Viby J, Aarhus, Denmark) was utilized as a process-directing agent and added to the calcium solution at a concentration of 50 µg/mL. Equal volumes of calcium and phosphate solution were then mixed, and the prepared collagen scaffolds were placed on a stainless steel wire mesh that was immersed in the mineralization solution (300 mL for each sample). Solutions were agitated using a stir bar, at 300 rpm, placed below the wire mesh. Samples were mineralized for 3 days in an incubator at 37 °C, then removed from the mineralization solution and rinsed of soluble salts by placing the mesh sample basket in a 500 mL beaker with DI H_2_O and stirring for 45 min, with the DI H_2_O replaced every 15 min.

### 2.4. Crosslink Density Evaluation

The approximate crosslink density was evaluated through an assay that quantifies the free amine group content of collagen scaffolds. EDC-crosslinked collagen samples, and methacrylated collagen samples before UV-crosslinking (~2 mg each), were soaked in 0.5 mL of a 4% (*w*/*v*) NaHCO_3_ solution, then 0.5 mL of a 0.5% (*w*/*v*) 2, 4, 6-trinitrobenzenesulfonic acid (TNBS) solution was added. The mixture was heated at 40 °C for 2 h, then 1.5 mL of 6 M HCl was added. The samples were then heated at 60 °C for 90 min, then cooled to room temperature. The solutions were aliquoted into wells of a 96-well plate in triplicate, and absorbance was measured using a microplate reader at 415 nm. A blank was prepared using the same procedure, but without a collagen sample in the solution. The absorbance values were compared to a non-crosslinked collagen control.

### 2.5. SEM Imaging and Analysis

Mineralized collagen samples were lyophilized, and a portion of each sample was mounted onto an SEM stub, which was then double carbon coated for SEM analysis. Each scaffold was imaged in a TESCAN MIRA3 SEM (Tescan, as., Brno, Czech Republic) equipped with an EDAX Octane Pro SDD energy dispersive spectrometer (EDS) at an accelerating voltage of 5 kV. EDS was performed at an accelerating voltage of 15 kV to confirm the presence of calcium and phosphate peaks from the mineral, and to assess their relative amounts with respect to the carbon peak from organic matrix. Pseudo-quantitative elemental analysis was performed from characteristic X-ray intensities to obtain Ca and P atomic % values using EDAX TEAM™ software (Pleasanton, CA, USA). Analysis of fibril morphology and diameter was performed using the ImageJ software (NIH, Bethesda, MD, USA) with 15 fibrils measured on triplicate samples for each sample condition.

### 2.6. Thermogravimetric Analysis

Thermogravimetric analysis was utilized to assess the degree of mineralization in each composite sample. Samples were heated from 30–800 °C at a heating rate of 20 °C min^−1^. Samples were heated in an air environment to maximize combustion of all organics, so that the material remaining at 700 °C could be used to evaluate the mineral content after all the organic portion of the samples had been combusted.

### 2.7. Mechanical Testing

Tensile testing measurements of mineralized crosslinked collagen samples were performed on an Instron 5943 universal testing system. The tensile tests were conducted on scaffolds in the wet state, immediately after removal from DI H_2_O, at a rate of 2 mm/min at room temperature. Samples were immersed in water for 1 h after the mineralization and rinsing process described earlier, immediately before loading into the tensile testing grips to prevent sample dehydration during testing. The thickness, width, and testing length of each sample (rectangular strips with dimensions around 2 cm × 0.5 cm × 100 µm strips) was measured with a caliper micrometer before testing to determine tensile stress and strain values that were then used for the calculation of tensile strength and modulus during testing. Tensile strength at break, % elongation, and Young’s modulus were calculated by the Instron software (Bluehill 3, Norwood, MA, USA) and confirmed by analysis of the raw data. All tensile testing was performed on samples in triplicate (*n* = 3 for each degree of crosslinking within each method), and statistical analysis of the measurements was performed using the IBM SPSS Statistical Software (Version 27.0. Armonk, NY, USA: IBM Corp). One-way ANOVA was utilized to analyze the effects of the degree of crosslinking on the modulus and tensile strength of non-mineralized and mineralized collagen scaffolds (α = 0.05). Tukey’s HSD test for multiple comparisons was used to evaluate differences between groups for each variable.

## 3. Results

### 3.1. Evaluation of Crosslinking Density

The density of crosslinks in the EDC-NHS crosslinked samples was measured through a TNBS assay, where the reaction of TNBS with primary amines can be measured through UV–Vis absorbance. The crosslink densities calculated from these absorbance readings ranged from 4 to 31% based on the reduction in absorbance compared to the non-crosslinked collagen controls (Table 1). Because the methacrylated collagen is prepared through the reaction of primary amines on collagen groups, a TNBS assay was also performed on the methacrylated collagen prior to crosslinking, to determine its extent of modification; assuming that all the acrylate groups will be involved in crosslinks, the absorbance value of 0.253 obtained from the UV–Vis measurements corresponds to an approximate degree of modification of 16% for the 300 s timepoint, relative to the unmodified TeloCol^®^, which is similar to the 1:1 crosslinking density measured for EDC at the 1:1 conditions.

### 3.2. Mineralized Fibril Morphology

Fibril morphology and diameter were analyzed from SEM images of crosslinked and non-crosslinked mineralized fibrils. SEM images of all the mineralized collagen scaffolds show plump fibrils, generally a sign that the fibrils are well infiltrated with mineral, in contrast to non-mineralized matrices where the fibrils collapse and flatten upon drying. As expected for the simple densification process used here, there is a random organization of fibril orientation for all samples. This appearance and organization remained fairly consistent among samples for all crosslinking conditions and intensities (Figure 1). Values for fibril diameters from each crosslinking condition and intensity are recorded in Table 2 and Appendix A (see in Appendix A). Samples in the EDC, RF, and quercetin crosslinking groups showed only small changes in fibril diameter, with average diameters ranging between 192 and 231 nm. The samples crosslinked with EDC appeared to show a slight decrease in fibril diameter with increased crosslinker concentration, except at the highest level. The riboflavin-incorporated samples appeared to show a slight increase with crosslinking times. The methacrylated collagen samples possessed a much larger fibril diameter, with an average value over 300 nm for the non-crosslinked and 45 s crosslinked samples, but this decreased for the 90 s and 300 s crosslinking conditions to values closer to those of the other crosslinker groups. Fibrils also appeared to be partially fused in some regions of the larger diameter groups (Figure 1E,F), so this may account for the significantly larger fibrils for the shorter crosslinking times. One might speculate that the added hydrophobicity of the methacrylate groups might contribute to the partial fusion during fibrillogenesis, although it is not clear why this was reduced for the higher crosslinking exposure times, unless the acrylate groups react with something other than another collagen chain, to form polar pendant side groups.

### 3.3. Mineral Content

Although the presence of plumped up fibrils is suggestive of intrafibrillar mineral, one cannot see the internal mineral with SEM imaging, so EDS analysis was also utilized to confirm mineral infiltration into the collagen fibril structures. Calcium and phosphate mineral content was determined from EDS analysis of each mineralized collagen sample, and representative spectra are shown in the Appendix A (Appendix A, see in Appendix A), along with calcium:phosphate and calcium:carbon peak ratios provided in Appendix A (see in Appendix A), which provides a semiquantitative method of comparing the relative degrees of mineralization of the matrices. The Ca/P ratios were in the range of 1.51–1.59, which is similar to bone with respect to being calcium-deficient hydroxyapatite. Samples across all groups displayed similar levels of Ca content, ranging from 25–30 wt% (Appendix A, see in Appendix A), similar to (and even higher than) the values of ~25 wt% previously measured in native bone specimens [36]. In the EDC-crosslinked group, Ca content appeared to slightly decrease with an increase in crosslink density, and changes in mineral content were not significant in the quercetin group. Riboflavin and methacrylated collagen groups showed a decrease in mineral content from non-crosslinked to crosslinked groups, but then showed a slight increase as crosslinking density increased.

The mineralized collagen samples were assessed using TGA to more quantitatively compare the degree of mineralization for each respective crosslinking density. This provides an assessment of the bulk mineral content, as compared to the surface content probed by EDS. Mineral content values were determined from mass% remaining at 700 °C, after all organic material had been combusted, and are shown in Figure 2, with specific values listed in Table 2. TGA graphs for each set of samples are shown in the Appendix A, Appendix A (see in Appendix A). EDC-crosslinked samples showed the greatest difference in degree of mineralization as crosslinking density increased, decreasing from 77% in the non-crosslinked samples to 51% in the 1:1 and 2:1 EDC:collagen crosslinked groups. This was surprising given that the EDS peak ratios of Ca/C and wt% Ca (Appendix A, see in Appendix A) exhibited higher values of mineral content than the other crosslinkers, suggesting that the mineral is more concentrated at the surface of the scaffolds, given that EDS only measures microns in depth at the surface. The methacrylated samples and quercetin-crosslinked samples displayed degrees of mineralization ranging from 73–77 wt%, while samples treated with riboflavin solutions (before crosslinking and mineralization) possessed slightly lower degrees of mineralization, from 69–74 wt%.

### 3.4. Tensile Testing

The modulus and ultimate tensile strength of the mineralized collagen samples were obtained from micro-tensile testing measurements performed in the wet state (Figure 3, Appendix A, see in Appendix A). Different trends in Young’s modulus and tensile strength were observed between the various crosslinking methods. Samples in the EDC- and RF-crosslinking groups displayed markedly higher values of tensile strength and modulus as compared to the quercetin and methacrylated collagen groups. For both the EDC and RF groups, there was a large jump in both properties from the non-crosslinked to crosslinked samples, although the trends with crosslinking density beyond that initial jump differed. EDC showed a steady and significant increase in strength, while RF’s strength decreased at the higher crosslinking value (*p* < 0.01 for all crosslinked groups). Surprisingly, the modulus decreased for both EDC and RF crosslinking after the initial jump, although both still retained a higher modulus than the non-crosslinked samples. For the RF-crosslinked group, both tensile strength and modulus were maximized in samples exposed for 1 min to UV irradiation (Figure 3). This was followed by a decline in both properties at 300 s, and with UV exposure greater than 5 min, shrinking and/or warping of the thin collagen films occurred, so samples were not able to be accurately tested. Tensile strength was very low for all the quercetin and methacrylated samples, and neither showed a significant jump between non-crosslinked to crosslinked samples, even though there was a modest but significant jump in modulus. This, however, was not followed by any further increases in modulus. These two groups possessed relatively low tensile strengths as compared to the EDC- and RF-crosslinked samples, regardless of crosslinking density.

Non-crosslinked and crosslinked samples were also tested in the non-mineralized state, to assess the impact of crosslinking alone on the mechanical behavior of the collagen matrix itself (Figure 4, Appendix A, see in Appendix A). The quercetin and methacrylated collagen samples were tested at their highest respective crosslink density, due to the minimal changes in mechanical properties observed in tests of the mineralized samples across crosslinked groups. The quercetin and methacrylated collagen samples were not able to be tested in the non-crosslinked state, as the gels did not hold together in the clamps of the tensile testing machine before tests began. Apparently these two modified collagens are inherently weaker, and this cannot be effectively overcome by crosslinking. Quercetin samples crosslinked under UV irradiation for 300 s were robust enough for testing, but displayed much lower values of modulus and tensile strength than any of the other crosslinked collagen samples. EDC and riboflavin groups were tested at all crosslinking intensities that were used for mineralized samples. For the EDC-crosslinked group, both modulus and tensile strength increased with increasing crosslinking density up to 1:1, but then exhibited a decline at the 2:1 ratio, which was particularly pronounced for the tensile strength. Riboflavin-crosslinked samples showed a similar increase in both properties up to 1 min of UV irradiation, but then exhibited a decline at 300 s.

## 4. Discussion

Densified fibrillar type I collagen scaffolds were synthesized through the concentration of collagen solutions after fibrillogenesis, as described in previous studies [27,30,31]. The scaffolds were modified using EDC-NHS crosslinking, quercetin modification, or UV-mediated crosslinking of methacrylated and riboflavin-incorporated collagen, to assess the effect of each method on mineral infiltration and the mechanical properties of the respective scaffolds with various crosslinking intensities. Non-crosslinked and crosslinked scaffolds were mineralized through a PILP process directed by OPN, which resulted in intrafibrillar mineralization of the collagen for all the methods of crosslinking. Tensile testing was utilized in this study to assess the bulk behavior of the fibrillar collagen scaffolds, adding to previous cantilever-based AFM indentation [30] and nanomechanical data [27] from our group on similar scaffolds. Tensile testing allows for the assessment of interactions between mineralized fibrils and fibers in the collagen scaffold. For all crosslinkers, there was an increase in modulus upon mineralization, which was expected given that the fibrils become embedded with mineral nanocrystals. However, there was a drop in the tensile strength for many samples going from the non-mineralized (Figure 4) to mineralized (Figure 3) conditions, which is discussed below.

The EDC-NHS and riboflavin crosslinkers yielded the most promising improvements in mechanical properties. There was an increase in both stiffness and tensile strength of EDC-NHS crosslinked scaffolds over the non-crosslinked samples for the non-mineralized conditions (Figure 4), which is likely due to the amide bonds created between the carboxylic acid and amine groups of collagen side chains [17]. These bonds serve to increase the resistance of collagen fibers to tensile stresses and failure, as has been observed in non-mineralized constructs [18], and the increases in tensile strength in this study suggest that this effect persists after the structures have been mineralized (Figure 3). However, while tensile strength and modulus display a dramatic increase from non-crosslinked to crosslinked samples, the modulus values declined after this initial jump, even though the tensile strengths continued to increase consistently at the higher ratios. A decrease in fibril diameter (Table 2) may suggest that intrafibrillar crosslinks are formed in addition to interfibrillar crosslinks, limiting the infiltration of the mineral precursor that normally swells fibrils during mineralization, but not to an extent that is too detrimental to the mechanical properties of the scaffold. In fact, it may be beneficial in keeping the fibrils from becoming overly infiltrated with the more brittle mineral phase. This is also reflected in the TGA measurements (Table 2), which confirm the decrease in mineral content with crosslinking density (dropping from 77% to 65% with the lowest EDC dose, and then further to 51% at increasing EDC levels). Thus, while the decreasing mineral contents may lead to the reduced moduli, the correspondingly larger collagen contents may provide the necessary interconnectivity of the organic matrix to contribute to their enhanced strengths. Given the opposing trends of modulus and strength beyond the initial jump, the mid-range properties at the 0.5:1 and ~1:1 ratios of EDC to carboxylate groups on collagen seem to provide the optimal combination of mechanical properties. Interestingly, these conditions led to mineral contents that happen to fall in the range of bone (60–65 wt%) and antlers (45–59 wt%) [37,38], where either modulus or strength are differently optimized for the different requirements of bone (skeletal stiffness) versus antler (fracture resistance during battle).

An inverse trend of reduction in strength with high levels of mineralization is seen for all the crosslinkers (Figure 5), consistent with previous observations by our group, where more robust collagen samples (e.g., strong turkey tendon) displayed brittle and fragile behavior when mineralized through the PILP processes (unpublished results). In the case of collagen samples previously fabricated by our group, very high values of mineral content (up to 80%) were reached in PILP mineralized samples [30]. Though at the time we were striving for maximal intrafibrillar mineralization to be achieved, the excessively high degree of mineral content found in most of our samples may be above the percolation threshold, resulting in a lower interconnectivity of the collagen matrix, diminishing the effect of this natural polymer in resisting tensile failure due to various stresses. Collagen is ‘designed’ for carrying tensile loads, which is especially apparent in soft tissues, but also known to be the case for bone [39]. In the EDC-crosslinking system, the lower degrees of mineralization (~50–65 wt%) resulted in this resistance to tensile failure being more successfully retained. This demonstrates the importance of maintaining the structure of an interpenetrating nanocomposite [2,40,41]. One must also keep in mind that the biomimetic PILP process uses only one simple protein mimic, rather than providing the complex assortment of NCPs that are found in bone formation, which likely optimizes the amount of intra- versus inter-fibrillar mineral that is found in native bone. Perhaps more importantly, the enzymatic crosslinking that occurs in situ during the collagen assembly of osteoid tissue is difficult to emulate. It may be that biological crosslinking provides intrafibrillar bonds, that regulate the amount of mineral precursor that is able to infiltrate the fibrils.

Exposure of riboflavin to UV causes the generation of reactive oxygen species, which react and form crosslinks with various collagen residues, including arginine, histidine, and lysine [42]. These crosslinks are more varied than those of EDC or methacrylate-crosslinked collagen samples, so the crosslinking density could not be directly compared to other methods. However, other reports on soft tissue constructs have shown riboflavin to provide comparable tensile strengths to EDC-NHS-crosslinked scaffolds [17], which was the case here for the lower level of EDC (0.5:1). However, while higher levels of EDC crosslinking in the non-mineralized samples led to further increases in strength, there was an eventual decline at the highest ratio, and this effect appeared to have already reached a maximum for riboflavin at the 60 s exposure, which had comparable strength and modulus to the 0.5:1 EDC-crosslinked samples (Figure 4). A similar trend was seen in the mineralized films, with tensile strength and modulus being highest in the 60 s riboflavin-crosslinked scaffolds. However, while the higher level of riboflavin crosslinking led to an even higher modulus value, the strength was reduced, and both levels of riboflavin crosslinking gave significantly lower strengths than the EDC system, which may once again result from a loss of collagen interconnectivity from the high degrees of mineralization (70–74 wt%, Table 2) that occurred with riboflavin crosslinking. The detrimental effects of excessive UV exposure are also present in this method, as exposure had to be limited to 300 s due to matrix warping past that time, and even at this 300 s timepoint, the mechanical properties have declined. 

Quercetin [20] modifies collagen constructs through formation of non-covalent bonds, which seemed to result in smaller changes in tensile strength and modulus of non-mineralized scaffolds, with values just a little above the zero crosslinking conditions of the other crosslinkers. In mineralized scaffolds, both modulus and tensile strength displayed no significant change with crosslinking density, and values for both properties remained lower than those of most other groups tested. This lack of change in properties is possibly due to the weaker intermolecular interactions created by quercetin, or the disruption of these interactions during collagen infiltration with high degrees of mineral.

Methacrylation of collagen for UV crosslinking modifies amine groups, primarily on lysine residues of the collagen. Exposure to UV light in the presence of an Irgacure photoinitiator creates bonds between these acrylated amine groups. In previous studies of soft collagen gels, an increase in exposure time has been shown to result in an increased crosslink density, leading to an increase in the stiffness of collagen constructs when tested in compression [43]. The bonding between methacrylate groups on collagen modified for UV crosslinking was expected to have a similar effect to the amide bonds formed by EDC-NHS, with respect to mechanical properties, but our results show no significant difference between the mechanical properties as UV exposure time is increased for the methacrylated samples (Figure 3), unlike for the EDC group. This was also true for the non-mineralized group (Figure 4), so it may be that all the acrylate groups have already reacted to their fullest extent. However, given that most of the non-mineralized methacrylated samples were too weak to even perform tensile tests, and the strength and modulus values were much lower than the non-crosslinked controls in the other groups (Figure 4), there seems to be something disrupting the properties. The lack of improvement in tensile strength may be an effect of the methacrylation process of the collagen before processing into scaffolds, where methacrylate groups replace amine side groups and may add hydrophobicity to the collagen. This differs from the EDC and RF methods, which crosslink and modify fibrils after scaffold formation. SEM image analysis of the methacrylated collagen scaffolds before crosslinking (Figure 1E) shows that the fibrils have a somewhat different morphology, being larger in diameter and more fused, than scaffolds that have not been modified by methacrylation (Figure 1A,I,M). This indicates that the methacrylation process affected fibril formation and organization in this study, which in turn likely led to the less robust mechanical response to tensile stresses. This seemed to create an inherently weaker collagen matrix, and was an effect that apparently could not be overcome with the addition of crosslinking or mineralization.

One complication in comparing the properties of biomimetic scaffolds to native bone is the fact that mechanical measurements of bone are highly variable, depending on many factors such as structural level, location, bone mineral content, and collagen fiber orientation [44]. For large specimens of dense cortical bone, moduli in the range of 14–22 GPa in the wet state are reported [45]. However, modulus values vary in specimens of smaller size. Testing of specimens in bending on a microstructural level has yielded values of 1.5–5.4 GPa in the wet state [46], with values also decreasing with specimen dimension from ~1 mm to 100 µm in size. This is even more significant at the nanostructural level. Our AFM nanoindentation studies of PILP-mineralized collagen films yielded modulus values of 130–180 MPa, and testing of bovine cortical bone under the same conditions resulted in a modulus value of 5.5 GPa, while other groups found even lower values, in the range of 1.79–2.70 GPa [27,30].

Our modulus and strength values are lower than secondary (lamellar) bone, as expected, but seem reasonable given the porosity of the collagen scaffolds, which are more similar to primary (woven) bone. Nanomechanical testing of woven bone (rat fracture callus) measured a modulus of ~11.2 GPa; however, this was in the dehydrated state [47], and it dropped to ~0.2 GPa in the hydrated state [48]. Our tests were performed in the hydrated state, so the crosslinked samples gave moduli that seem similar to, or better than, woven bone (the testing methods differ, so one cannot accurately compare). Further studies, that start with dense lamellar collagen constructs, are needed to determine if properties closer to secondary bone might be achieved; but we assume the overall trends will be applicable to other collagen processing methods, such as the promising ‘molecular crowding’ approach to produce liquid-crystalline collagen organization, which yields a dense cholesteric microstructure that closely resembles the twisted plywood microstructure of lamellar bone [30].

## 5. Conclusions

The collagen matrices synthesized in this study are large enough to measure properties beyond the microscale and probe interactions between fibrils, but not as large as the samples typically used to measure tensile strength and modulus in many bulk samples. The modulus values of ~3.5 GPa recorded in this research are still comparable to the 1–5 GPa values recorded in native bone samples with similar sample dimensions, even though the method of testing was different [46]. EDC-NHS crosslinking and UV exposure with riboflavin incorporation proved successful in improving the mechanical properties of densified isotropic collagen scaffolds. The effects of the EDC-NHS crosslinking seem to be twofold, both increasing the strength and modulus of the collagen films themselves, and apparently limiting the amount of mineral infiltration into the matrix, allowing the tensile properties of the collagen matrix to be retained in the interpenetrating composite material. Improvements in strength might be anticipated for the riboflavin crosslinker (and others) by carefully controlling the degree of mineralization, to maintain collagen interconnectivity. Further enhancements could be anticipated when using the “molecular crowding” approach to more closely mimic the density and organization of lamellar bone. We believe that combining the “molecular crowding” approach to mimic osteoid formation, with the biomimetic PILP processing for intrafibrillar mineralization, has the potential to create dense, tunable, bone-like scaffolds that mimic both the resorbable nanostructure of mineralized collagen fibrils and the load-bearing lamellar microstructure of secondary bone, leading us closer to the goal of creating the next generation bone substitute, that is able to sustain loads while being remodeled by the bone remodeling unit.

## Figures and Tables

**Figure 1 polymers-15-01981-f001:**
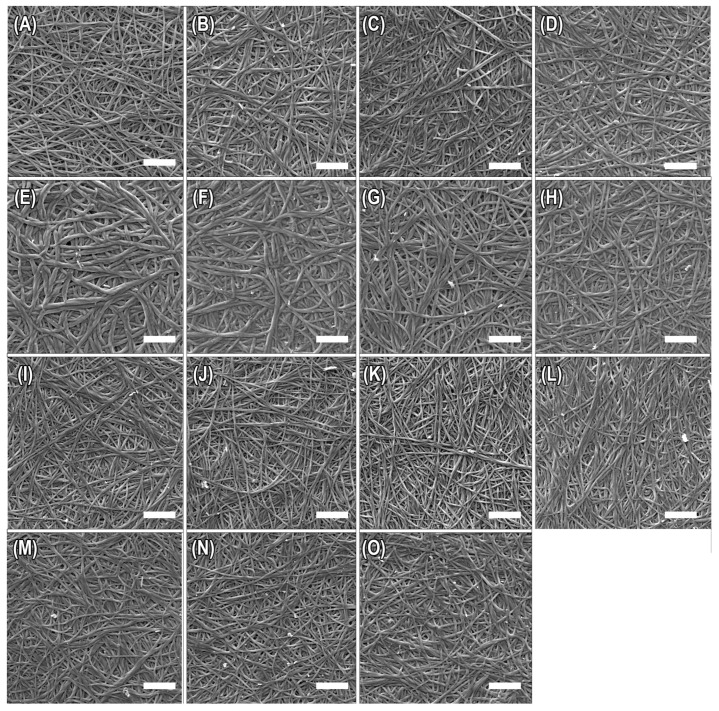
SEM micrographs of mineralized collagen, comparing four types of crosslinkers. Row 1 depicts quercetin-modified collagen, with quercetin-collagen w/w ratios of (**A**) 0:1 (non-crosslinked), (**B**) 0.5:1 (50%), (**C**) 1:1 (100%), and (**D**) 2:1 (200%). Row 2 depicts methacrylated collagen crosslinked by UV exposures of (**E**) 0 s (non-crosslinked), (**F**) 45 s, (**G**) 90 s, and (**H**) 300 s. Fibrils appear noticeably larger in diameter for the lower levels of this crosslinker. Row 3 depicts collagen crosslinked by EDC-NHS at EDC:collagen w/w ratios of (**I**) 0:1 (non-crosslinked), (**J**) 0.5:1, (**K**) 1:1, and (**L**) 2:1. Row 4 depicts riboflavin-incorporated collagen crosslinked by UV exposure for (**M**) 0 min (non-crosslinked), (**N**) 1 min, and (**O**) 5 min. Scale bars = 5 µm.

**Figure 2 polymers-15-01981-f002:**
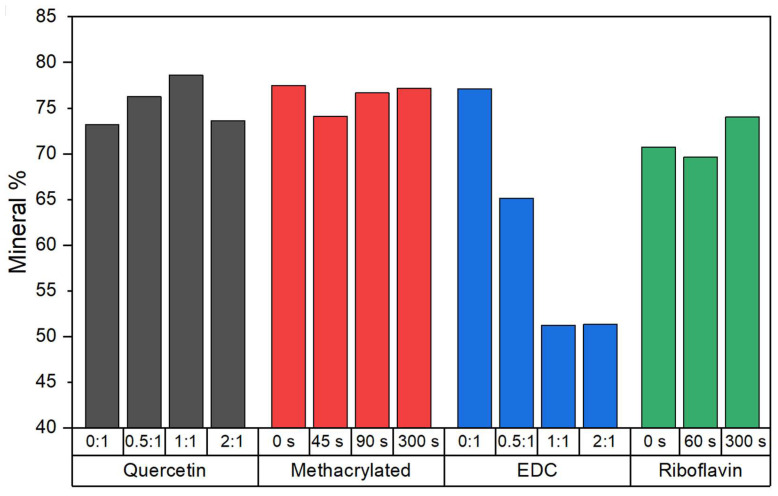
Mineral content of mineralized collagen composites, comparing four types of crosslinkers. Mineral wt% was determined from TGA mass% remaining at 700 °C, after all organic material had been removed by combustion.

**Figure 3 polymers-15-01981-f003:**
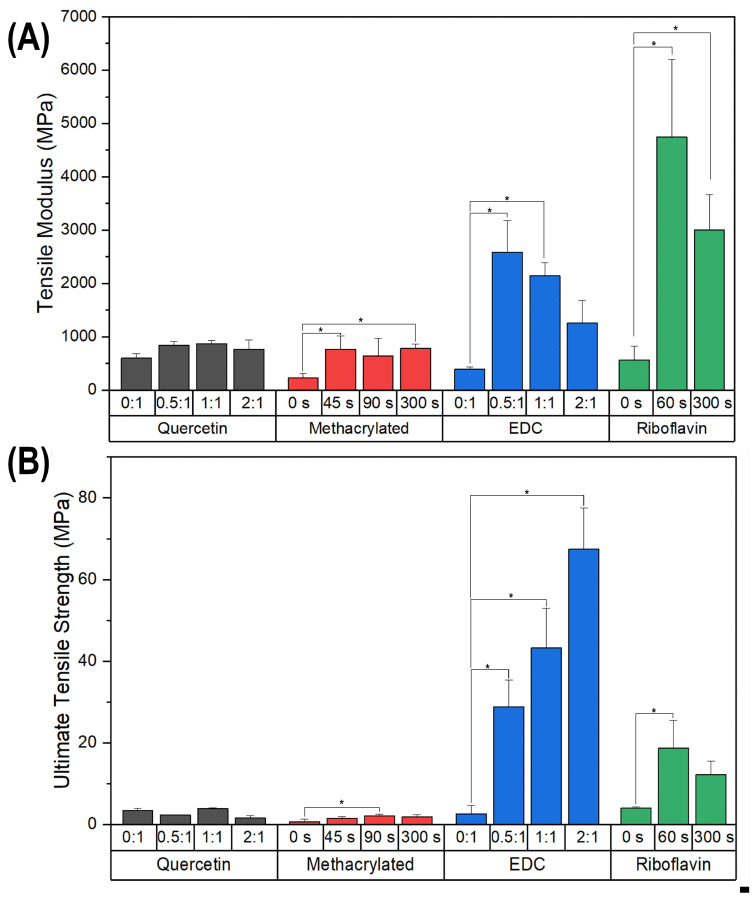
(**A**) Modulus and (**B**) tensile strength measurements across all crosslinking groups for mineralized samples, based on respective degrees of crosslinking for each type. Methacrylated and quercetin samples displayed a smaller change in modulus with crosslinking, relative to EDC- and RF-crosslinked samples. * denotes mean differences significant at the 0.05 level.

**Figure 4 polymers-15-01981-f004:**
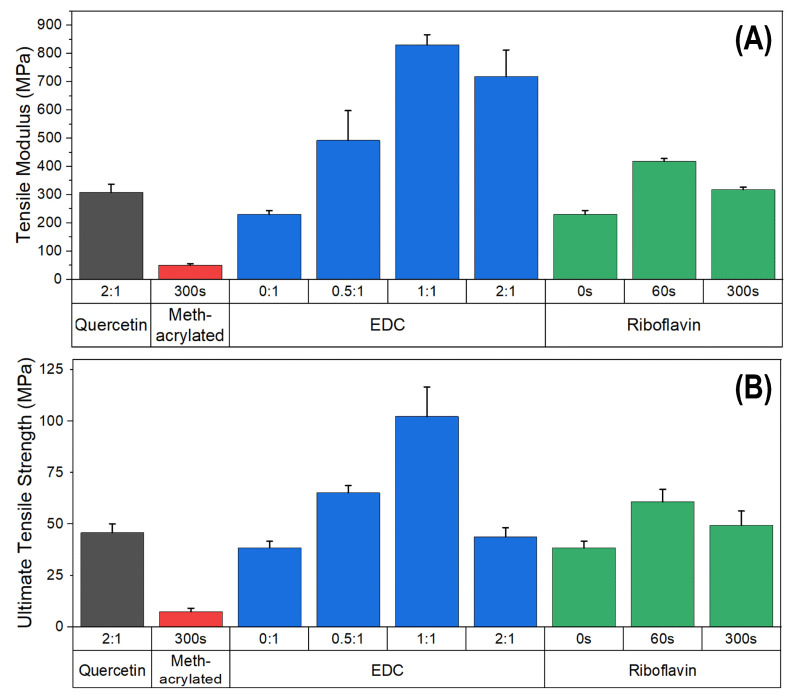
(**A**) Modulus and (**B**) tensile strength measurements across all crosslinking groups for non-mineralized samples, based on respective degrees of crosslinking for each type. Crosslinked quercetin samples displayed slightly higher values than the non-crosslinked samples of EDC and riboflavin, but a significantly lower modulus relative to EDC- and RF-crosslinked samples. The only methacrylated sample that could even be measured, exhibited much lower values than all other samples, with properties deteriorated below even the non-crosslinked samples of EDC and riboflavin.

**Figure 5 polymers-15-01981-f005:**
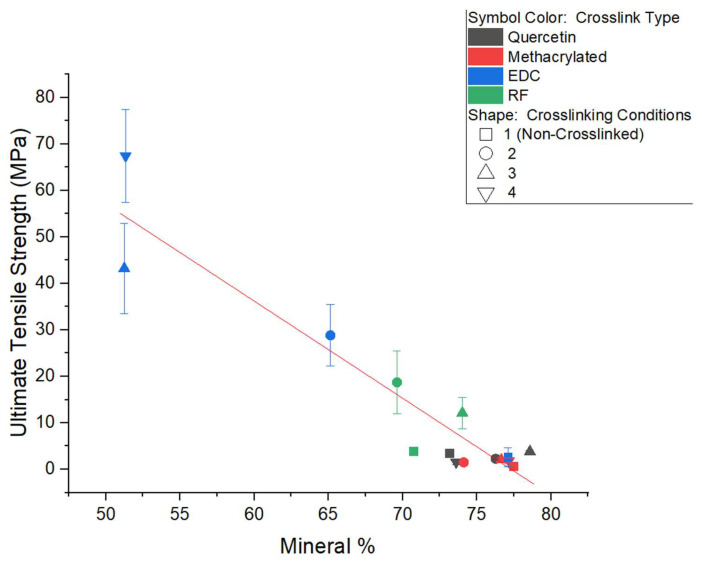
Relationship between mineral content (wt%) and tensile strength for all mineralized scaffolds. Tensile strength tends to decrease with increased mineral content in crosslinked samples.

**Table 1 polymers-15-01981-t001:** Approximate crosslink density from TNBS assay absorption measurements. Sample # denotes crosslinking condition, with 1 being the non-crosslinked samples.

EDC Sample #	EDC:COO^−^	Absorbance (415 nm)	Crosslinking Density (%)
1	0:1	0.30	0.00
2	0.5:1	0.29	4.6
3	1:1	0.25	16.9
4	2:1	0.21	31.4

**Table 2 polymers-15-01981-t002:** Thermogravimetric analysis data and fibril diameter measurements for mineralized collagen scaffolds at each crosslinking condition. The first condition for each crosslinker is the negative control, without crosslinking. Mass % remaining after all organic material had combusted at 700 °C was used for wt% mineral determination.

Crosslinking Type	Crosslinking Conditions	Wt% Mineral	Average Fibril Diameter (nm)
Quercetin	0:1 *w*/*w*	73.2	200 ± 20
	0.5:1	76.3	209 ± 11
	1:1	78.6	231 ± 17
	2:1	73.6	208 ± 20
Methacrylated	0 s	77.5	328 ± 60
	45 s	74.1	348 ± 43
	90 s	76.7	233 ± 35
	300 s	77.2	262 ± 27
EDC	0:1 *w*/*w*	77.1	211 ± 18
	0.5:1	65.1	202 ± 18
	1:1	51.2	192 ± 15
	2:1	51.3	214 ± 23
Riboflavin	0 s	70.7	225 ± 35
	60 s	69.6	220 ± 20
	300 s	74	215 ± 37

## Data Availability

Not applicable.

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
