# Peer review of "Influence of Crosslinking Methods on Biomimetically Mineralized Collagen Matrices for Bone-like Biomaterials"

_polymers, 2023, doi:10.3390/polym15091981_

Round 1

Reviewer 1 Report

The manuscript is an interesting scientific work. The results can be recommended for practical research.

In the experimental part, the characteristics of collagen are not given. Basic information can be obtained on the website https://advancedbiomatrix.com/telocol3.html . However, not all data are presented there, for example, there are no molecular weight characteristics that can influence the gelation process. In addition, it was desirable to indicate why bovine collagen was used for research.

Question to the authors, are there any studies confirming the possibility of practical application, for example, the results of the MTT test? Such results would allow a fuller assessment of the practical significance of the work.

The comments are of the nature of recommendations and do not reduce the overall impression of reading the manuscript.

Reviewer 2 Report

The effect of different crosslinking molecules (EDC-NHS, quercetin, methacrylate collagen, and riboflavin) on the degree of mineralization via PILP (using OPN) method, tensile strength, and modulus of hydroxyapatite-mineralized scaffolds were analyzed and compared. Improved modulus and tensile strength values were obtained using EDC-NHS and riboflavin crosslinking methods.

1. Is commercial collagen I disassembled in primary molecules or assembled into tropocollagen units before being submitted to fibrillogenesis?

2. In this work it is not clear to me how you analyze and distinguish intrafibrillar mineralization excluding extrafibrillar mineralization.  Are you sure the complete mineralization is intrafibrillar? Probably the presence of some extrafibrillar mineralization can influence the average fibril diameter and the mechanical properties.

3. Can you include the formula of the different crosslinking molecules?

Reviewer 3 Report

Abstract: Not complete, lack of study results

1. Introduction

Reduce this part, too much information, make concise intro.

 2. Materials and Methods

2.1. Collagen Scaffold Densification and Preparation: Dense collagen scaffolds were prepared...The densified collagen films in the dialysis... The scaffolds were then cut and: not clear, the samples were films or scaffold?

The authors just done fewer experiments (literally three SEM, TGA and Texture) and concluded the study. In order to confirm the Bone-Like Biomaterials, the authors should do some evidence study to justify the hypothesis like Bioactivity (using simulated body fluid), biocompatibility, and support in bone cells growth and so on. Right now, the study plan is not complete.

3. Results

EDC appeared to show a slight decrease in fibril diameter with increased: Present the result of fibril diameter here to confirm.  

Figure 2.: Incomplete. Provide SD, Error bar, statistics.

Figure 3. statistics

Concise the discussion 

Round 2

Reviewer 3 Report

The manuscript still exists the same issues, the authors did not revise the manuscript carefully based on my comments. The manuscript does not acceptable unless the authors properly sort-out the flaws in their manuscript based on my comments. 

Further comments

1. Introduction Reduce this part, too much information, make concise intro. We provided information that we considered important to understanding the reasons for the study and methods chosen to study. It’s not clear what parts the reviewer considers too much information.:

New comment 1: Its up to the authors where to reduce, since this part covers almost 2 and half page, which make the reader tedious.

 2.1. Collagen Scaffold Densification and Preparation: Dense collagen scaffolds were prepared...The densified collagen films in the dialysis... The scaffolds were then cut and: not clear, the samples were films or scaffold? The methods section has been updated to provide clarity on the process and nature of the collagen constructs, the abstract and figures have also been updated to clarify results and significance. 

New comment 2: Still the problem exists. (Refer Lines no 162, 169, 171 and 175.  collagen scaffolds or collagen films?

Figure 2.: Incomplete. Provide SD, Error bar, statistics. The bar graph is given to summarize data from the TGA curves shown in the supplemental data. Because of the nature of these curves and the small size and initial weight of the scaffolds, they are generally not presented with statistical analysis, but the origin of the presented data has been clarified.

New comments 3: Not convincing, Did the authors repeat the experiment (Mineral wt. % from TGA mass %) with three independent experiment set-up? If not, then the authors should repeat this experiment at-least three times to get technical replicate and provide the SD value. Axis label was blurred. 

New comments 4: Figure 1: Images A, C, F, M, N and O were more contrast than others. Homogenize the images with uniform background. Also, all the images were blurred, provide a good resolution image. Now the differences were not visible.

New comments 5: Figure 3.: Poor image quality, the legends and axis were unreadable. 

New comments 6: Figure 4. Provide a good image with at-least better image resolution, not acceptable in present form. 

New comments 7: Figure S6: The same problem, SD, Error bar, statistics

New comments 8: Why the supplementary tables were presented in between Fig S6 and Fig.S7.

The authors should carefully revise the manuscript based on my comments this time. 

Round 3

Reviewer 3 Report

No more comments.